# EG-SIF: Improving Appearance Based Gaze Estimation using Self Improving Features

**Vasudev Singh**[*1]                              VASUDEV.SINGH@MERCEDES-BENZ.COM
**Chaitanya Langde**[*†1,2]                        CHAITANYA.LANGDE@GMAIL.COM
**Sourav Lakhotia**[1]                             SOURAV.LAKHOTIA@MERCEDES-BENZ.COM
**Vignesh Kannan**[1]                              VIGNESH.V.KANNAN@MERCEDES-BENZ.COM
**Shuaib Ahmed**[1]                                SHUAIB.AHMED@MERCEDES-BENZ.COM
[1] *Mercedes-Benz R&D India*
[2] *Indian Institute of Technology Bombay*

## Abstract

Accurate gaze estimation is integral to a myriad of applications, from augmented reality to non-verbal communication analysis. However, the performance of gaze estimation models is often compromised by adverse conditions such as poor lighting, artifacts, low-resolution imagery, etc. To counter these challenges, we introduce the eye gaze estimation with self-improving features (EG-SIF) method, a novel approach that enhances model robustness and performance in suboptimal conditions. The EG-SIF method innovatively segregates eye images by quality, synthesizing pairs of high-quality and corresponding degraded images. It leverages a multitask training paradigm that emphasizes image enhancement through reconstruction from impaired versions. This strategy is not only pioneering in the realm of data segregation based on image quality but also introduces a transformative multitask framework that integrates image enhancement as an auxiliary task. We implement adaptive binning and mixed regression with intermediate supervision to refine capability of our model further. Empirical evidence demonstrates that our EG-SIF method significantly reduces the angular error in gaze estimation on challenging datasets such as MPIIGaze, improving from 4.64° to 4.53°, and on RTGene, from 7.44° to 7.41°, thereby setting a new benchmark in the field. Our contributions lay the foundation for future eye appearance based gaze estimation models that can operate reliably despite the presence of image quality adversities.

**Keywords:** Eyegaze, Adabins, Appearance based, Image quality, Multitask.

## 1. Introduction

Eye gaze is a fundamental non-verbal communication cue that encapsulates many insights about human intent, shaping the landscape of numerous applications across diverse domains such as human-computer interaction (Zhang et al., 2019; Li et al., 2019; Wang et al., 2015), driver monitoring system, reading analysis (Beymer and Russell, 2005), screening for dyslexia, augmented reality (Patney et al., 2016), etc. and hence the need of estimating the eye gaze accurately.

Appearance-based methods using convolutional neural networks (CNNs) that directly estimates human gaze from images captured by cameras are the most commonly used as

---

[*] Equal contribution

[†] Work done as intern at Mercedes-Benz R&D India

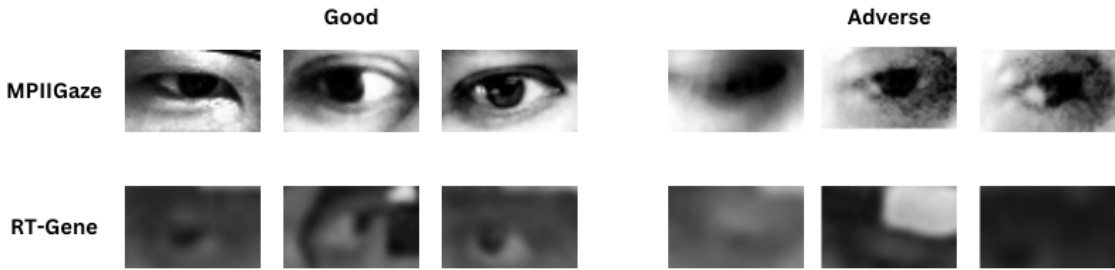

Figure 1: Segregated good and adverse images in MPIIGaze and RT-Gene dataset.

they provide better gaze estimation performance (Murthy and Biswas, 2021; Cheng et al., 2020). From images, gaze can be estimated through full face or head of a subject or only through eye images. In applications, where considering privacy risks are important, working on face images for training the model is difficult. Moreover, in applications like driver monitoring, eye images are preferred over face images as it needs real-time estimation on the edge devices with low computational capability.

Nevertheless, the presence of inherent adverse factors, such as inadequate lighting conditions, subjects' head movements away from the camera during eye image acquisition, etc. results in the introduction of noise and compromised resolution in certain images within the dataset. We hypothesize that datasets like MPIIGaze (Zhang et al., 2017) and RT-GENE (Fischer et al., 2018) contain a subset of images affected by these adversities, significantly impacting the accuracy of gaze estimation. Therefore, it becomes imperative for gaze estimation methods to robustly identify and address these challenges.

In order to model a robust CNN architecture, we propose an eye gaze estimation with self improving features (EG-SIF) method for learning noise independent features. Our methods consists of three main parts: 1. Segregating the dataset into two disjoint sets i.e., good images that are relatively free from adverse effects and the adverse quality images. 2. A transformation that generates adverse quality image given an image from the good set by transferring the noise distribution of the image from the adverse set. 3. A multitask end-to-end training framework with image enhancement as an auxiliary task along with gaze estimation to make network robust towards noise. This type of segregation of varying quality images and training them in different way has been accomplished for the first time in this paper. With the proposed method,[1] it is observed that model is able to perform better on adverse quality images and also able to surpass the current state-of-the-art methods.

The main contributions of this paper are summarized in the following:

- We propose a novel framework for gaze estimation that can operate reliably even in case of image quality adversities.

- Data segregation into two disjoint sets based on quality of images in the dataset for the first time.

---

1. https://github.com/vasu-dev/EG_SIF/

- A novel end to end training framework with image enhancement at the core as an auxiliary task to make the network robust to noise.

- Adaptive binning based mixed regression with intermediate supervision for the first time in gaze estimation.

- With the above proposed methods we are able to obtain the state-of-the-art performance on benchmark datasets that lay the foundation for future eye appearance based gaze estimation models.

## 2. Related work

**Gaze estimation**: Traditional methodologies predominantly focus on the identification and analysis of ocular movement patterns, encompassing phenomena such as fixations, saccades, and smooth pursuits. In contrast, model-based approaches concentrate on the extraction of geometric attributes like pupil centre, contours, and eye corners (Park et al., 2018). With the advancements in deep learning techniques and the abundance of data, appearance-based methods have emerged as a dominant paradigm. They focus on learning a non-linear mapping between the eye image and the associated gaze (Tan et al., 2002). Appearance-based methods can be further divided on the basis of the input provided: eye images or full-face images.

*Features from eye images*: Early deep-learning methods (Zhang et al., 2015) provided a single-eye greyscale low-resolution image in conjunction with head pose information to estimate the gaze from the features. As computational resources burgeoned, deeper networks (Zhang et al., 2017) based on a vanilla VGG-16 or ResNet, were designed to push the boundaries of gaze estimation accuracy. Subsequently, it was found that concatenating features from two eyes yielded better accuracy (Fischer et al., 2018), after which even a four-stream network (Cheng et al., 2018) was built to extract features. (Chen and Shi, 2019) uses dilated convolutions as a means to extract high-level eye features, which efficiently increases the receptive field size of the convolutional filters without reducing spatial resolution.

*Features from face images*: In the realm of gaze estimation, researchers have recognized the potential richness of information contained within facial images, which encompass details such as head pose and offer higher-resolution representations compared to individual eye images and thus recently more attention has moved towards face image-based gaze estimation. (Krafka et al., 2016) was one of the first attempts where the face image along with the left and right eye images were used to estimate gaze. Various facial and eye detectors are used to crop out the region of interest. In (Abdelrahman et al., 2022) they divide the gaze range into discrete bins converting the regression problem to a classification problem alongside running independent networks for different components of 2D gaze (yaw and pitch). Gaze-360 (Kellnhofer et al., 2019) used a pinball loss function to predict error quantiles, indicating confidence in the prediction. However, studies found that just concatenating the features is insufficient and attention-based mechanisms came into the picture. (Cheng et al., 2020) argues that the weights of two eye features are determined by face images due to their specific task, so they assign weights with the guidance of facial features. Recently (Murthy and Biswas, 2021) used an attention branch in parallel to feature extraction to eventually attain weighted eye features and to be used with the face features. More recently

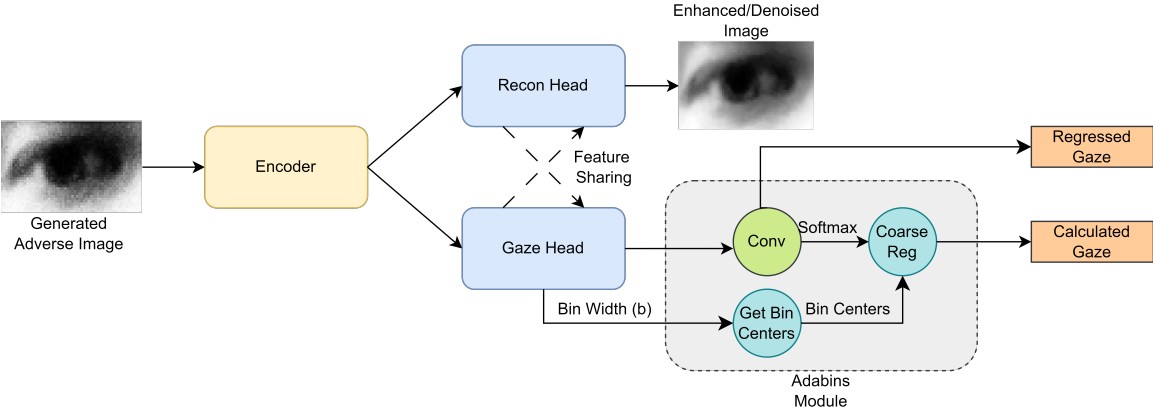

Figure 2: Our Proposed Network Architecture. The network consists of four main components, the encoder block, image enhancement head, the gaze head and a adabins module. The input to the network is a normalized eye crop of spatial dimension H x W and the output is the 3D Gaze (Yaw and Pitch).

transformer-based models (Cheng and Lu, 2021; Yu et al., 2021) have also been applied to test their efficacy in attending to features and generating diverse features. To the best of our knowledge, we are the first ones to utilise deep supervision to aggregate multi-layer features for supervision.

Attempts have been made to utilise additional information like eye landmark features (Wu et al., 2019), and pupil centre (Lee et al., 2020) to improve gaze estimation accuracy. But eye appearance varies much across different people thus making the task of cross-person testing extremely difficult. To solve this either calibrated methods are used or invariant features are obtained via the network. (Park et al., 2018) convert the original eye images into a unified gaze representation, which is a pictorial representation of the eyeball, the iris and the pupil and further regress the gaze from the representation. FAZE (Park et al., 2019) uses an autoencoder to learn the compact latent representation of gaze, head pose and appearance. They introduce a geometric constraint on gaze representations, i.e., the rotation matrix between the two given images transforms the gaze representation of one image to another. Further, they train a highly adaptable gaze estimation network through meta-learning. The network can be converted into a person-specific network once training with target person samples. Nowadays more attention has moved towards gaze redirection (via rotation equivariance (Jin et al., 2023) or NeRFs (Ruzzi et al., 2023)), synthesis and scene-based understanding of gaze.

**Image enhancement:** Data-driven methods are largely categorized into two branches, namely CNN-based and GAN-based methods. The nature of architecture is not of much interest in the context of this work, but the variation in loss functions plays a crucial role. (Zhao et al., 2017) discusses the effectiveness of using a mixed loss which accommodates both structural similarity-based losses and simple error functions ($l_1$ and $l_2$)

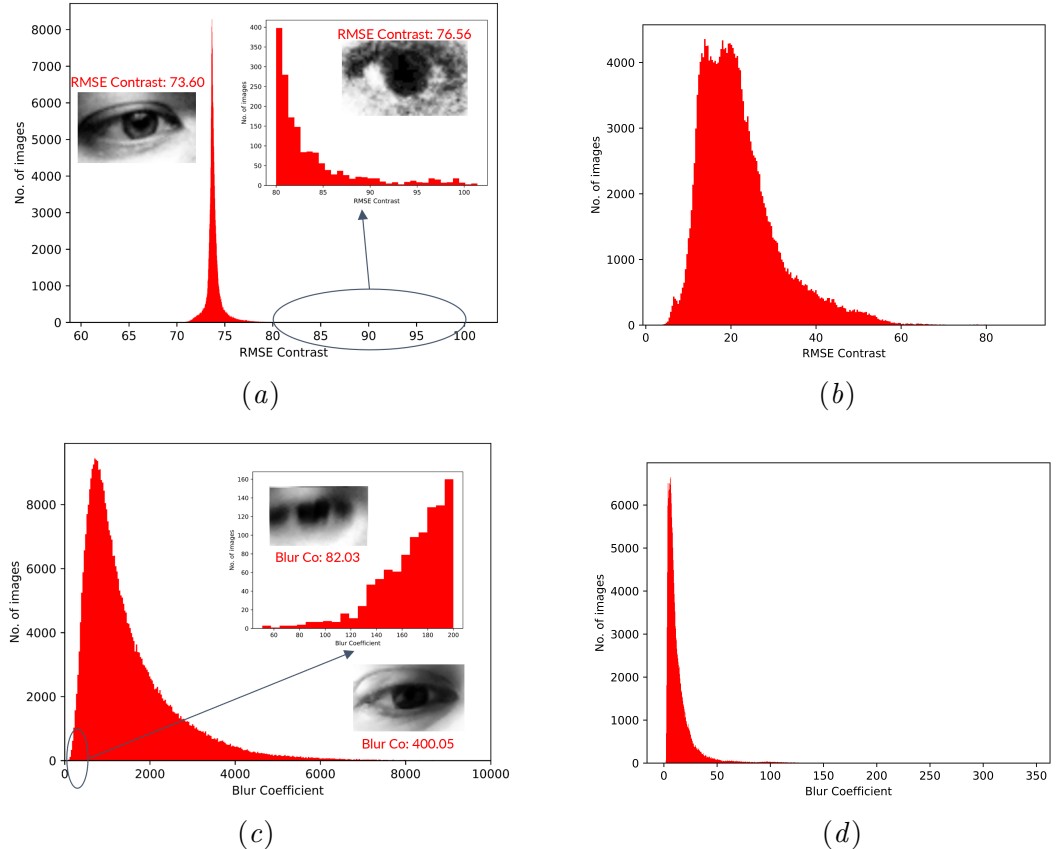

Figure 3: (a) RMSE contrast distribution for MPIIGaze, higlighting the long tail and corresponding images (b) RMSE contrast distribution for RTGene (c) Blur coefficient distribution for MPIIGaze, highlighting low blur coefficient area along with corresponding images(d) Blur coefficient distribution for RTGene.

## 3. Gaze estimation using self improving features

In this section, we discuss the details of gaze estimation using self improving features. It consists of 3 parts as discussed in detail in the following subsections:

### 3.1. Image segregation

The first step is to identify and segregate the entire training set $I$ into two disjoint subsets $S$ and $F$, where $s_i \in S$ are images with lower adverse effects and of high quality and $f_i \in F$ are with poor quality. For this, we did some preliminary analysis on the image properties namely RMSE contrast and blur co-efficient on both the datasets, namely MPIIGaze and RT-Gene. The intuition behind this is that eye images with high constrast indicate noise and high blur leads to loss of essential information needed for gaze estimation. The results of this study are depicted in the Figure 3. As it can be seen in the Figure, there is significant number of images in the long tail of datasets in terms of both RMSE and blurriness.

Based on the above analysis, the segregation function $y : I \rightarrow \{s, f\}$ that maps each image to its respective category is defined as:

$$y_i = \begin{cases} f, & \text{if RMSE-contrast}(i) > \lambda_r \text{ and Blur-coefficient}(i) < \lambda_c \\ s, & \text{otherwise} \end{cases} \tag{1}$$

where,

$$Blur - coefficient(i) = Variance(Laplacian(i)) \tag{2}$$

and

$$RMSE - contrast(i) = \sqrt{(p_j - \mu)^2/n} \tag{3}$$

whereas, $p_j$ is the intensity of $j^{th}$ pixel of image $i$, $\mu$ is the mean intensity value of $i$ and $n$ is the total number of pixels in $i$.

It is observed that high RMSE-contrast and low blur-coefficient values resulted in a noisy images. In case of MPIIGaze, we observed that images with RMSE Constrast $> 75$ (Fig. 3($a$)) and Blur coefficient $< 200$ (Fig. 3($c$)) fall into the subset $F$, hence, $\lambda_r$ is fixed at 75 and $\lambda_c$ at 200. whereas, for RT-Gene $\lambda_r$ at 10 (Fig.3($b$)) and $\lambda_c$ at 10 (Fig. 3($d$)). With careful analysis of the baseline models on these segregation, we found that most of them perform poorly on adverse images. As it can be seen in Table 1, in both MPIIGaze and RT-Gene dataset the angular error trained on baseline (ResNet-18) for adverse images is significantly higher than the overall average.

| Dataset | Total | Good | Adverse |
|---------|-------|------|---------|
| MPIIGaze | 5.59 | 5.53 | 5.79 |
| RT-Gene | 8.9 | 8.83 | 9.61 |

Table 1: Angular errors on average, good, and adverse subsets of the datasets using baseline.

To test that the segregation of good and adverse is accurate, a comparative study was done assessing the entropies of the subsets as shown in Table 2. Inferences were run on 3 pre-trained models with similar architectures and the uncertainty in gaze prediction was used as the basis to calculate the entropy of an image. The analysis revealed a lateral positive shift in the distribution of the entropies of adverse images as compared to the good images.

### 3.2. Good and adverse pair generation

Once the training set $I$ is segregated into the two disjoint subsets, $I = S \cup F$ and $S \cap F = \Phi$, we use the images from $S$ and the noise distribution from set $F$ through histogram matching $H(f, s)$ to generate adverse samples (similar to image in set $F$) from $S$. We take a random pair of images $(s_j \in S, f_i \in F)$ from both the sets and then generate a adverse quality image as a linear combination of the matched histogram $H(f_i, s_j)$, that transfers the pixel

| Data split | Joint entropy |
|:----------:|:-------------:|
| Good | $6.2 \pm 2.1$ |
| Adverse | $7.3 \pm 2.4$ |

Table 2: Joint entropy calculated on the segregated good and adverse set for MPIIGaze using mean $\pm$ standard deviation of 3 different runs

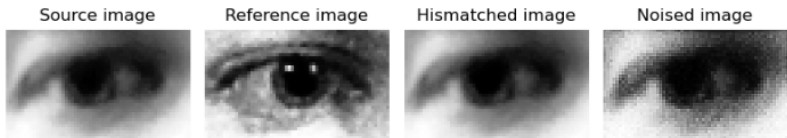

Figure 4: Conversion of a source image into a noisy image using the properties of a reference image as well as salt-n-pepper noise.

distribution, salt-n-pepper noise ($N_s$) for high contrast generation and Gaussian noise ($N_g$) for blur effect as given in the equation below:

$$f'_{ij} = H(f_i, s_j) + \alpha N_s + \beta N_g \qquad (4)$$

$$\text{where, } s_j \in S; \ f_i \in F; \ \alpha, \beta \in [0, 1]$$

Now we have a pair($f'_{ij}, s_j$) of good image($s_j$) and a corresponding generated adverse image($f'_{ij}$) with the same gaze direction but added noise. This acts as a training pair for image denoising/enhancement. Fig. 4 shows the transition of a good image to a adverse image. As it can be noted that the transformed good images are poor in quality.

### 3.3. Multitask network architecture

It has been observed in numerous studies (Park et al., 2019) that gaze estimation accuracy suffers significantly during cross-person testing, thus the architecture is primarily designed to be robust to such settings. The proposed EG-SIF network draws inspiration from the proven attributes of the U-Net (Ronneberger et al., 2015) architecture, including the encoder's proficiency in capturing multi-scale contextual information, the utility of skip connections for preserving fine-grained details of learned features and the robustness towards noise. We modify it into a multi-task network with 2 task-specific decoder heads focusing on gaze estimation and image enhancement/denoising (refer to Fig. 2). The rationale underlying this multi-task approach lies in the intuition that regressing gaze coordinates from a denoised image is more tractable, thus the reconstruction head forces the encoder to learn self-improving features which effectively serve as proxy features of a denoised image. These information-dense features from the encoder are subsequently fed into the gaze head to predict 3D gaze. Following the ideology of (Abdelrahman et al., 2022) we model the horizontal(yaw) and vertical(pitch) components separately by using independent regressors or fully connected layers. This allows the network to capture the unique characteristics of each component independently. They also suggests that this improves the learning of the

network as it now has two signals that backpropagate. Additionally, we use the following elements to increase the robustness of the network:

1. **Cross-parameter sharing:** To provide both high-level spatial features and deep features for gaze estimation, parameter sharing across the heads inspired from Wang et al. (2020) was utilised (refer to Algorithm 1 for details). This ensures multi-layer information sharing and also encourages the generalisation of the network as it promotes the network to learn common features and representations that are useful for both tasks. Additionally, parameter sharing augments the network's capacity to learn noise independent features that possess meaningful attributes across distinct scales or layers of the architecture.

---

**Algorithm 1** Cross-feature sharing

---

$y$ : (Features from Gaze Head)
$x$ : (Features from Reconstruction Head)
$z = Concat(x, y)$
$i_g = F_3(y, z)$     (Intermediate Gaze Features)
$i_r = F_4(x, z)$     (Intermediate Reconstruction Features)
$z_1 = Concat(x, i_g)$
$z_2 = Concat(y, i_r)$
Where: $F_i$ represents a simple CNN block and in case of intermediate features, help in cross-fusion, and $z_i$ are the latent features which further propagate into the decoders

---

2. **Adabins:** Initially proposed by (Abdelrahman et al., 2022), gaze bin classification converts the regression task into a simpler classification task which can serve as a coarse estimation of the final gaze. However, using fixed predefined bins might not prove effective for the datasets in consideration since most of the gaze directions are distributed near the origin and have a sparse population near the ends. Thus we propose to use Adaptive bins (Adabins) for gaze bin classification inspired by their recent success in depth estimation (Bhat et al., 2021). Adaptive binning allows for the dynamic adjustment of bin boundaries based on the characteristics of the gaze data. This adaptability ensures that bins are optimized to capture gaze patterns effectively for a given dataset or user, enhancing the accuracy of gaze estimation. Additionally, Adaptive binning can create bins that are finer in regions with high gaze density and coarser in regions with sparse gaze data (refer to Fig. 5). Once the bins are obtained, the bin centres (b(c)) are calculated and scaled accordingly.

$$b(c_i) = g_{\min} + (g_{\max} - g_{\min}) \left( b_i + \frac{b_{i+1} - b_i}{2} \right) \tag{5}$$

Where: $g_{max}$ and $g_{min}$ are the minimum and maximum gaze values corresponding to the dataset, and $b_i$ is the predicted bin width

A parallel, fully connected layer (x) is then utilized for generating predictions associated with each bin. These predictions are subjected to a softmax activation function, resulting in the assignment of probability scores to each individual bin. A coarse gaze

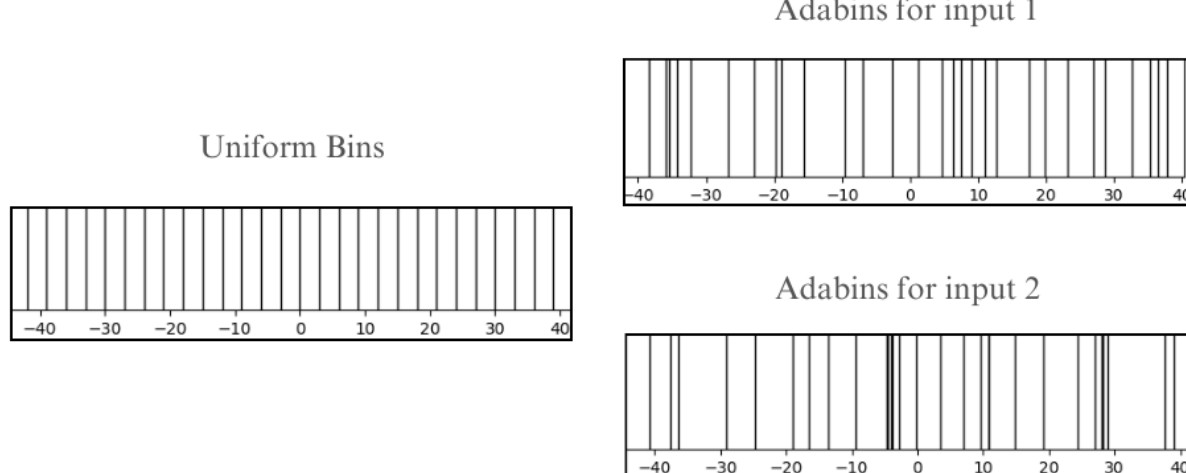

Figure 5: Example showing comparison b/w uniform and adaptive bins, as can be seen from the figure using adaptive bins can lead to better results as the bins are adapted to each input image individually.

is calculated as the expectation of the bin centres. The final gaze is calculated by adding a multi-layered perceptron (MLP) on top of the predictions regressing a single value.

$$G_{coarse} = \sum_{i=1}^{N} b(c_i) \cdot \text{Softmax}(x_i) \tag{6}$$

### 3.3.1. Loss functions

Many Convolutional Neural Network (CNN)-based models for gaze estimation commonly output predictions in the form of 3D gaze direction angles, typically represented in spherical coordinates as yaw and pitch. They frequently employ the mean-squared error ($l_2$ loss) for penalizing their networks. We propose the following loss function for training the EG-SIF network:

1. **Reconstruction loss:** Based on (Zhao et al., 2017) we use the suggested mixed loss. The mixed loss allows one to strike a balance between pixel-level accuracy (sharpness) and perceptual quality. $l_2$ loss (MSE) alone tends to produce overly smooth images because it primarily penalizes pixel-wise differences. The inclusion of multi-scale SSIM(Structural Similarity) helps mitigate this issue by encouraging the preservation of structural and textural details. The overall reconstruction loss is:

$$L_{\text{R}} = \lambda L_{\text{MSE}} + (1 - \lambda) L_{\text{SSIM}} \tag{7}$$

Where, $\lambda$ is a weight factor

2. **Gaze-estimation loss:** We utilize both regression and classification losses along with a regularization loss on the gaze heads.

- *Coarse and direct gaze-estimation loss:* The standard MSE loss is used to penalise both the directly regressed gaze and the calculated coarse gaze.

$$MSE(y, p) = \frac{1}{N} \sum_{i=1}^{N} (y_i - p_i)^2 \qquad (8)$$

Whereas $p_i$ is the regressed or calculated gaze and $y_i$ is the ground truth value

- *Gaze classification loss:* To penalise the predicted bin class, the standard cross-entropy loss is used.

$$H(y, p) = -\sum_i y_i \log p_i \qquad (9)$$

Whereas, $p_i$ is the predicted class vector and $y_i$ is the ground truth one hot class label

- *Bin-width regularisation:* This novel loss acts as a form of regularization, discouraging extreme values of bin widths. This can help prevent over-fitting and improve the generalization performance of a model.

$$\text{LimitLoss}(LL) = \max\left(0, \max(b(c_i)) - \text{weight} \cdot \text{threshold}\right) \qquad (10)$$

Whereas, the threshold is set as the validation score of the previous epoch

The overall gaze-estimation loss can be represented as:

$$L_G = \alpha \cdot (LL + H(y, p)) + \beta \cdot MSE_{coarse}(y, p) + MSE_{direct}(y, p) \qquad (11)$$

Where: *alpha* and *beta* are hyper parameters for loss-weighting

Thus the final loss for the network becomes:

$$L_{EG-SIF} = \rho L_G + (1 - \rho) L_R \qquad (12)$$

Where: $\rho$ is again a hyperparameter

## 4. Experiments and results

### 4.1. Datasets

With the development of appearance-based gaze estimation methods, many large-scale datasets with variations in gaze direction, head pose and appearance have been proposed. In order to assess the network, we train and evaluate our model on two popular datasets: MPIIGaze (Zhang et al., 2017) and RT-Gene dataset(Fischer et al., 2018).

**MPIIGaze** is the most popular dataset for gaze-estimation methods. It contains a total of 213,659 images (for each eye) obtained from 15 different subjects, collected over several months thus providing variations in illumination. The dataset contains normalised

eye images for corresponding face images, which is of our main interest. It also consists of a standard evaluation set which comprises 3,000 images (1,500 left-eye and 1,500 right-eye images) for each subject. The standard testing practice for the MPIIGaze dataset is the leave-one-out setting to check the model accuracy in a cross-person fashion.

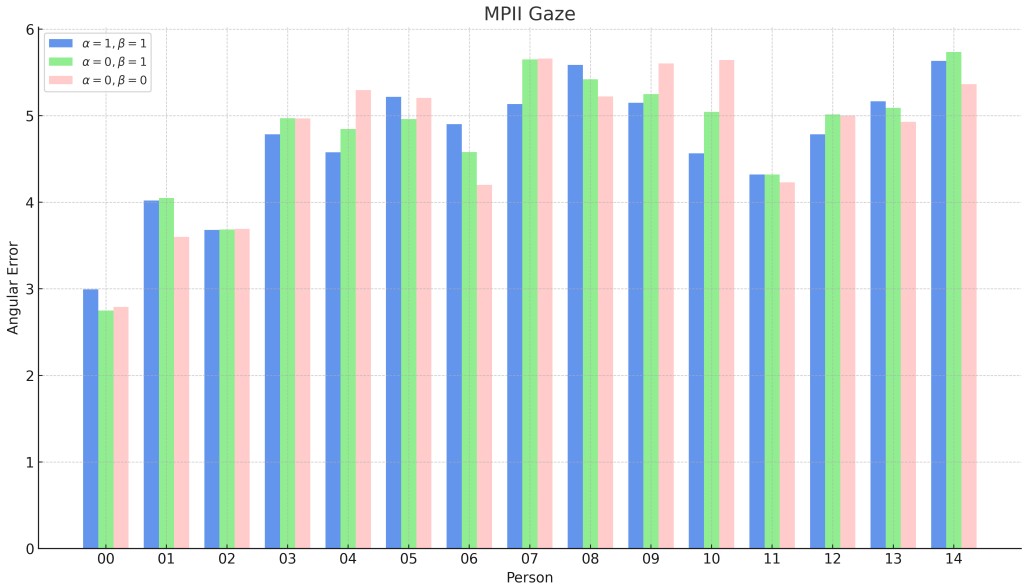

Figure 6: Results of using different hyperparamters($\alpha$, $\beta$) for MPIIGaze.

**RT-Gene** dataset was collected in a controlled laboratory environment, ensuring high-quality data. The dataset includes eye-tracking data obtained from participants using head-mounted eye-tracking glasses resulting in a much higher variation in the gaze angle distribution. It contains 122,531 inpainted and original images (229,116 distinct eye crops) acquired from 15 distinct subjects. Since the inpainted images are heavily noised images as reported by (Murthy and Biswas, 2021), we use original images for training and testing. The standard testing practice for RT-Gene is using a 3-fold cross-validation.

### 4.2. Data pre-processing

We use normalised eye images of 36x60 resolution for both datasets and do not make use of any other additional information (e.g. full-face image, head pose vector) to obtain the results. The ground truth class labels for the images were calculated during training. As a result, the dataset had both class labels and continuous gaze labels making them suitable for our proposed overall loss function. A total of 30,966 adverse images were obtained in the MPIIGaze dataset leaving 396,352 training and validation samples. In the case of the RT-Gene dataset, 18,971 adverse samples and 103,650 training and validation samples were used respectively. The generated good-adverse pairs are such that the adverse samples of a certain subject inherit noise properties from all the available subjects making the

| Network architecture | Angular error |
|---|---|
| Baseline (Resnet-18) | $5.6 \pm 0.9$ |
| Reconstruction | $5.31 \pm 0.52$ |
| Reconstruction + cross-param sharing | $5.06 \pm 0.77$ |

Table 3: Angular errors on model trained using the generated adverse dataset demonstrating the effectiveness of the mixed-loss and cross-parameter sharing on MPIIGaze. The results reported are mean $\pm$ standard deviation in evaluating cross persons.

dataset more exhaustive. Another modification was to horizontally invert right eye images to align geometrically similar features with left eye, thus making it easier to train a model. Horizontally inverting left eye images without inverting right eye images is strongly expected to yield similar results.

### 4.3. Training and results

We performed leave-one-out cross-validation on the MPIIGaze dataset using the proposed models. The entire dataset apart from the held-out set was used for training and the held-out set was used for validation. All the experiments were done using a yytorch framework utilising adam optimiser. With a simple grid search initial learning rate was fixed at 0.001 and a multi-step decay with a decay constant at 0.5. Dropout was used for fully connected layers with a dropout probability of 0.25. We train the network for 25 epochs each using a batch size of 512. Loss weights($\alpha$, $\beta$) were changed and the performance was monitored and compared with the state-of-the-art. Downsampling and cross-parameter sharing were done twice in the network. For all the conducted experiments the hyperparameters were fixed at $\rho = 0.5$, $\lambda = 0.5$.

We utilize gaze angular error ($\theta$) as the evaluation metric following most gaze estimation methods. Assuming the ground-truth gaze direction is $\mathbf{g} \in \mathbb{R}^3$ and the predicted gaze vector is $\hat{\mathbf{g}} \in \mathbb{R}^3$, the gaze angular error ($\theta$) can be computed as:

$$\theta = \arccos \left( \frac{\mathbf{g} \cdot \hat{\mathbf{g}}}{\|\mathbf{g}\| \cdot \|\hat{\mathbf{g}}\|} \right) \tag{13}$$

The initial experiments were undertaken to evaluate the comparative performance of an encoder-gaze regressor network in contrast to an encoder equipped with two distinct decoders—one for gaze estimation and the other for image enhancement/denoising (Table 3). The enhancement based experiment was additionally constrained structural similarity loss along with the MSE. Notably, the introduction of cross-parameter sharing mechanisms yielded a significant enhancement in the network's performance, indicating the strong correlation between the tasks.

In the context of cross-person evaluation, using proper augmentations while training plays a pivotal role in enhancing the overall robustness of neural networks. The base augmentations used were brightness (0.5,1.5), contrast (0.5, 1.5), saturation (0.5, 1.5), hue (-0.1,

| Augmentation type | Angular error |
|---|---|
| Normal | 4.9 ± 0.83 |
| Patch | 4.82 ± 0.64 |
| Patch + SP noise | 4.79 ± 0.48 |

Table 4: Angular errors on model trained using the generated adverse dataset demonstrating the effectiveness of various augmentations on MPIIGaze. The results reported are mean ± standard deviation in evaluating cross persons.

0.1).These adjustments were applied in a randomized manner with a combined probability of 50%, imparting stochastic variability to the training data. Additionally guassian blur with a mean of 5 and std (1,3) with 50% probability was applied. Later we incorporated a customized patch augmentation technique. It helped the model focus on the pupil rather than the surroundings.The proposed patch augmentation applies white patches on the edge of an image with variable size at a variable position. This resulted in a slight increase in the accuracy of the model and a decrease in the variance. Another finding while visual inspection was that adding salt and pepper noise to the training images increases robustness towards noise in the image. Table 4 shows the respective improvements by using a combination of these augmentations.

Table 5 shows the effectiveness of adaptive bins(Adabins) over fixed bins. As proposed by Abdelrahman et al. (2022), we use 28 bins for the MPIIGaze and 42 bins for RT-Gene.

| Type of bins | Angular error |
|---|---|
| Fixed | 4.72 ± 0.52 |
| Adaptive (adabins) | 4.53 ± 0.46 |

Table 5: Angular errors on model trained using the generated adverse dataset demonstrating the effectiveness of Adabins on MPIIGaze. The results reported are mean ± standard deviation in evaluating cross persons.

Table 6 shows the comparison of our method with other methods, as it can be seen the proposed method EG-SIF produces significantly lower error on MPII dataset and slightly better performance on RT-Gene dataset. This is in comparison to the current state-of-the-art methods. Fig. 6 shows a comparative analysis of the effect of changing hyperparameters and subject-wise performance. From this we can observe that the hyperparameters are very sensitive to change in appearance and thus affect the average performance.

## 5. Conclusion

In this paper, we introduce, for the first time a method for eye gaze estimation with self-improving features (EG-SIF). As we observe that eye crop based dataset for gaze estimation

| Model | Angular error | |
|---|---|---|
| | **MPIIGaze** | **RTGene** |
| ARENet | 5.02° | - |
| RTGene | 4.8° | 8.6° |
| AGENet | 4.64° | 7.44° |
| **Ours(EG-SIF)** | **4.53°** | **7.41°** |

Table 6: Performance comparison with state-of-the-art gaze estimation methods.

contains both good and adverse images due to various inherent reasons and the current state-of-the-art methods do not explicitly focus on the image quality, hence compromising the performance. EG-SIF segregates images based on their quality. Once images are segregated, a transformation is defined to obtain adverse set of images from good images. This pair is used in the multi-task model where the task is to enhance/denoise the generated adverse image along with gaze estimation. Apart from this, methods including cross-parameter sharing and adabins are proposed to increase the efficiency. On evaluation of our methods, it is observed that it outperforms current state-of-the-art methods by a good margin. This concludes that adverse quality images in the dataset require separate treatment which can not only enhance the performance in the dataset but also during inference in real-time condition during those adversities.

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
