# OpenReview forum: "EG-SIF: Improving Appearance Based Gaze Estimation using Self Improving Features"
_NeurIPS.cc/2023/Workshop/Gaze_Meets_ML — Gaze Meets ML 2023 Oral_

### Official Review · Reviewer_vd4L · 2023-10-22
**Denoising using adverserial technique - improves gaze estimation in challenging (noisy) datasets**

**Rating:** 7
**Confidence:** 4

**Review:**

The study proposes a method for gaze estimation.  To train the model, the algorithm divides the dataset into high-quality and low-quality image subsets based on contrast and blurriness. Adverse samples are generated from the high-quality images using noise distributions from the low-quality subset. The adverse samples have been used for denoising. The system uses a combination of reconstruction loss and gaze estimation loss during the model training. The proposed approach is tested on two datasets which are mentioned as challenging dataset containing poor lighting and low-resolution images. Researchers reported comparison table which reports that the proposed approach outperforms existing methods for these challenging datasets.

The study`s technical contribution is employing a denoising method using generative adversarial technique which improves the gaze estimation.

There is a typo in Page 12 – in front of Table 6.

Table 6 – the metric definition is missing for this comparison table. Please include the metric either in the caption or in the text.

The term “Self-improving” in title may be confusing and could be interpreted as self-training algorithms. The main contribution here is denoising with adversarial method.

---

### Official Review · Reviewer_piZT · 2023-10-23
**Elegant implementation of intuitive idea**

**Rating:** 7
**Confidence:** 3

**Review:**

The authors present an approach for gaze estimation using for the first-time deep supervision to aggregate multi-layer/multi-resolution features for supervision that obtains clear superior performance when compared with the state-of-the-art methods in the MPIIGaze and RTGene datasets.

The presented approach first segregates images based on quality, generates pairs of good-adverse images, and then follows a UNet-based approach for multi-scale contextual information while leveraging skip connections for the fine-grained details. The UNet inspired approach introduces a cross-parameter sharing between the reconstruction and the gaze estimation head, and converts the regression problem to multi-class classification following an adaptive binning approach. The model convergence is determined based on an aggregate of loss functions and a regularization factor to avoid overfitting.
The idea once presented is intuitive, well-described, and seems well-implemented - albeit the link for the source code cannot be found.

Specific considerations for improvement:
- Fig 1: change the color of the histogram to green or red for improved visualization. Scale the x axis in all subplots to cover the extent of the apparent histogram, as currently it is difficult to tell where the tails end.
- Section 3.3: Interpretability is used as a buzz word nowadays. The authors should elaborate more on what they mean by the "interpretability of learned features."
- Section 4.2: Why specifically inverting the right eye?
- Section 4.2: Also this act with the right eye is counter-intuitive to me: when everything other step in the presented work is about improving the generalizability of the method, here the authors do something to harmonize the features. What is the benefit? Can the authors show an ablation study, where the eyes do not have similar features?
- Section 4.3.: How did the authors come up with these values for the learning rate? decay? dropout? Experimentation? literature? what is the effect of a change in these values in the overall performance?
- Section 4.3.: The authors claim that brightness, contrast, saturation, and hue augmentations were applied in a randomized manner, but do not mention with what probability. Please report towards facilitating reproduciblity.

Overall nice presentation, but I would expect the implementation to be provided in the form of source code to enable objective assessment.

---

### Official Review · Reviewer_2z33 · 2023-10-23
**Interesting idea - I recommend acceptance**

**Rating:** 7
**Confidence:** 3

**Review:**

Summary:
A method for gaze estimation is proposed. The main observation is that the quality of input images in the training data varies, and we need to account for this variation in the training process. Authors process to divide training data into two categories: "good" quality images and "adverse" quality images. This segregation is done heuristically based on two factors - contrast and blurriness. Now, a model is trained to do two tasks - (a) to transform adverse images into corresponding good images (ground truth for this is obtained using a hand-defined image enhancement model" and (b) gaze estimation. Experiments are done with different loss functions and architecture choices.

Strengths:
* Interesting idea to divide training dataset into groups according to the quality of images.
* This is the first time I am seeing image enhancement being used as an auxillary task.

Weaknesses (minor):
* Unclear how rmse-contrast and blurriness coefficient are computed.
* Imprecise writing and mathematical notation, such as in equation 2.

---

### Meta-Review · Area_Chair_F3Bo · 2023-10-26

**Recommendation:** Accept (Oral)
**Confidence:** 5

**Metareview:**

The paper presents a methodology for estimate eye gaze from eye images. The paper is well written and elegantly while providing comparative results. Reviewers suggested comments that could be addressed with relatively small effort where appropriate.

---

### Decision · Program_Chairs · 2023-10-26

Accept (Oral)